# CCDC88B is required for pathogenesis of inflammatory bowel disease

Nassima Fodil[1], Neda Moradin[1], Vicki Leung[1], Jean-Frederic Olivier[1], Irena Radovanovic [1], Thiviya Jeyakumar[1], Manuel Flores Molina[2], Ashley McFarquhar[1], Romain Cayrol[3], Dominique Bozec[4], Naglaa H. Shoukry[2], Michiaki Kubo[5], Julia Dimitrieva[6], Edouard Louis[6], Emilie Theatre[6], Stephanie Dahan[4,7], Yukihide Momozawa[5], Michel Georges[6], Garabet Yeretssian[4,8] & Philippe Gros[1]

Inflammatory bowel disease (IBD) involves interaction between host genetic factors and environmental triggers. CCDC88B maps within one IBD risk locus on human chromosome 11q13. Here we show that CCDC88B protein increases in the colon during intestinal injury, concomitant with an influx of CCDC88B[+]lymphoid and myeloid cells. Loss of Ccdc88b protects against DSS-induced colitis, with fewer pathological lesions and reduced intestinal inflammation in Ccdc88b-deficient mice. In a T cell transfer model of colitis, Ccdc88b mutant CD4[+] T cells do not induce colitis in immunocompromised hosts. Expression of human CCDC88B RNA and protein is higher in IBD patient colons than in control colon tissue. In human CD14[+] myeloid cells, CCDC88B is regulated by cis-acting variants. In a cohort of patients with Crohn's disease, CCDC88B expression correlates positively with disease risk. These findings suggest that CCDC88B has a critical function in colon inflammation and the pathogenesis of IBD.

[1] McGill Center for the Study of Complex Traits, Department of Human Genetics, Department of Biochemistry, McGill University, 3649 Sir William Osler Promenade, Montreal, QC, Canada H3G 0B1. [2] Centre de Recherche du Centre Hospitalier de l'Université de Montréal (CRCHUM), 900, Saint Denis Street, Pavillion R, Montreal, QC, Canada H20A9. [3] Département de Pathologie et de Biologie Cellulaire de l'Université de Montréal, C.P. 6128, succ. Centre-ville, Montréal, QC, Canada H3C 3J7. [4] Immunology Institute, Tisch Cancer Institute, Icahn School of Medicine at Mount Sinai, New York, NY 10029, USA. [5] Laboratory for Genotyping Development, Center for Integrative Medical Sciences, RIKEN 1-7-22 Suehiro-cho, Tsurumi-ku, Yokohama, Kanagawa 230-0045, Japan. [6] Unit of Animal Genomics, GIGA-R and Faculty of Veterinary Medicine, Universite de Liège (B34), avenue de l'Hôpital 1, 4000 Liège, Belgium. [7] Present address: SOBI, Inc, Waltham, MA 02452, USA. [8] Present address: The Leona M. and Harry B. Helmsley Charitable Trust, New York, NY 10169, USA. Correspondence and requests for materials should be addressed to N.F. (email: nassima.fodil@mcgill.ca) or to P.G. (email: philippe.gros@mcgill.ca)

I nflammatory bowel diseases (IBD) are heterogeneous inflammatory diseases that affect the gastrointestinal tract. In Crohn's disease (CD) the ileum and colon are primarily affected, whereas in ulcerative colitis (UC) only the distal colon is affected. Both genetic factors and environmental effects (life style, diet, intestinal flora) contribute to IBD pathogenesis. Genetic studies of patients with rare early onset and severe forms of IBD have uncovered >60 causative genes and associated mutations, and genome wide association studies (GWAS) have mapped >200 non-MHC linked loci that affect susceptibility to IBD[1–4].

However, the effect size and contribution to disease of individual GWAS loci is small; for many IBD loci the causative gene and mechanistic basis of the genetic effect are unknown.

In a forward genetic screen in mice, we identified that *Ccdc88b* is required for pathological and lethal neuroinflammation[5]. In mice, *Ccdc88b* mRNA transcripts are almost exclusively present in haematopoietic organs and the Ccdc88b protein is expressed in CD4[+] T cells, CD8[+] T cells and myeloid cell subsets[5]. *Ccdc88b* mutant (*Ccdc88b^{m1PGrs}*) T cells have impaired maturation in vivo, and reduced activation, cell division and cytokine production (IFNγ, TNF) in response to specific or non-specific stimuli in vitro[5]. CCDC88B belongs to the hook-related protein family (CCDC88A, CCDC88B, CCDC88C), which is defined by a conserved region similar to the microtubule binding domain of Hook proteins[6]. The function of these proteins is unclear, but they seem to functionally couple elements of the cytoskeleton with different cellular processes, such as vesicular transport and cell movement[6]. Ccdc88b has been shown to interact with the CDC42 guanine nucleotide exchange factor DOCK8[7]. Human *DOCK8* mutations cause primary immunodeficiencies associated with perturbed migration, altered function of myeloid and NK cells[8,9]. CCDC88B (HkRP3) is also required for NK cell cytotoxicity including production and mobilization of cytotoxic granules[8].

Human *CCDC88B* maps to distal chromosome 11 (11q13) within a locus associated with susceptibility to several inflammatory conditions[10], including sarcoidosis[11], IBD[1], psoriasis[12], alopecia areata[13], multiple sclerosis[14] and primary biliary cirrhosis[15]. The 11q13 locus contains >23 genes in linkage disequilibrium on a 1 Mb segment, making it difficult to identify the gene underlying the pleiotropic effect of this locus on inflammatory diseases. Epigenetic annotation based on recruitment, and transcriptional activation by proinflammatory factors IRF1, IRF8, and STAT1 in response to exposure of myeloid cells to IFNγ (myeloid inflammation score)[16], has been used to identify *CCDC88B* as the top "inflammatory" positional candidate at 11q13[5].

Here, we show that Ccdc88b[+] lymphoid and myeloid cells are recruited to the site of inflammation in experimental colitis. Furthermore, *Ccdc88b* mutant mice are protected against DSS-induced colitis, and naive *Ccdc88b* mutant CD4[+] T cells do not induce colitis in immunocompromised mice. In humans, *CCDC88B* mRNA and protein expression is increased in inflamed colons of patients with UC or CD. In human CD14[+] cells, *CCDC88B* mRNA is regulated by cis-acting regulatory SNPs (that is, eQTL), and eQTL effects and disease risk are correlated, with increased *CCDC88B* expression associated with increased risk. Our study therefore identifies a critical function of *CCDC88B* in colonic inflammation and IBD.

## Results

**CCDC88B expression is induced during experimental colitis.** The role of CCDC88B in intestinal homeostasis and in pathological inflammation was investigated in the dextran sodium sulfate (DSS) mouse model of intestinal colitis. We found that *Ccdc88b* mRNA levels gradually increased in the colon of DSS-treated wild-type (WT) mice at day 4 and day 8 following initiation of DSS treatment, when compared to untreated mice (Fig. 1a). Likewise, Ccdc88b protein level was increased at day 4 and day 8 post-treatment whereas no Ccdc88b expression was detected in the colon of *Ccdc88b* mutant mice at day 8 (Fig. 1b and Supplementary Fig. 7a). To investigate the tissue and cell types that express Ccdc88b in the colon during colitis, we performed immunohistochemistry and found Ccdc88b staining in a subpopulation of cells that are also positive for the hematopoietic marker CD45 (Fig. 1c). Interestingly, the rest of the E-cadherin positive intestinal mucosa and associated epithelium were negative for Ccdc88b (Fig. 1c). Furthermore, flow cytometry analysis (FACS) of mononuclear cells isolated from colons show a significant increase of CD45[+]Ccdc88b[+] cells at day 8 after DSS, confirming recruitment of Ccdc88b[+] inflammatory cells (Fig. 1d). Immunofluorescence and flow cytometry studies show that Ccdc88b[+] infiltrating cells belong to both the lymphoid (CD3[+], CD4[+], CD8[+]) and the myeloid (CD11b[+]) compartments (Fig. 1e, f). Further FACS analysis of infiltrating cells subsets identified increased recruitment of Ccdc88b[+] T cells, NK cells, neutrophils, inflammatory monocytes and macrophages at both day 4 and day 8 post DSS treatment (Supplementary Fig. 1). These findings indicate that during experimental colitis with DSS in mice, Ccdc88b[+] lymphoid and myeloid cells are recruited to the site of inflammation.

***Ccdc88b* mutant mice are protected from experimental colitis.** To directly assess whether CCDC88B plays a functional role in colitis, we subjected WT and *Ccdc88b* mutant (*Ccdc88b^{mut}*) mice to 3% DSS for 5 days followed by water for 3 days and evaluated colitis incidence (Fig. 2). *Ccdc88b^{mut}* mice showed significantly less body weight loss when compared to WT mice (Fig. 2a), and displayed significantly longer colons at day 8 post-treatment (Fig. 2b). Analysis of hematoxylin and eosin stained colon tissue sections revealed greater repair of the epithelial barrier and colonic crypts, thinner underlying muscle layer, and less abundant cellular infiltration in DSS-treated *Ccdc88b^{mut}* mice compared to controls (Fig. 2c). Histopathological assessment of a number of cellular and tissue phenotypes associated with colitis were quantified at day 4 and day 8 post-DSS treatment and confirmed reduced pathology and protection against DSS-induced colitis in *Ccdc88b^{mut}* mice (Fig. 2d). At both days 4 and 8, *Ccdc88b^{mut}* mice had lower infiltration of inflammatory cells, and reduced overall pathology measured by submucosal edema, gland loss and surface erosion and ulceration when compared to WT (Fig. 2e). No such differences in tissue architecture and histology were seen in untreated mice of either genotype (Fig. 2c).

Chemokines, such as MCP-1 and RANTES play critical roles in recruitment of pro-inflammatory cells to the site of lesions in

**Fig. 1** Mouse Ccdc88b expression in colon during DSS-induced colitis. Wild type (WT) mice were either not treated (NT) or given 3% DSS for 5 days followed by 3 days of water. **a** *Ccdc88b* mRNA expression in distal colons of NT ($n = 3$) or DSS-treated WT mice ($n = 3$ for each time point) at the indicated days. Data represent expression relative to *hprt*±SEM ($n = 3$). *$P < 0.05$, **$P < 0.01$ (two-tailed Student's *t*-test) and are representative of one experiment. **b** Representative immunoblots for Ccdc88b protein detected in extracts from distal colons from NT or DSS-treated WT and *Ccdc88B^{mut}* mutant mice at indicated time points, and densitometric quantification of Ccdc88b immunoblot normalized to β-actin±SEM ($n = 3$) (representative of one of two independent experiments); **$P < 0.01$ (two-tailed Student's *t*-test). **c** Representative confocal microscopy images of tissue sections from colon of NT or DSS-treated mice at indicated times, and stained with antibodies against Ccdc88b (red), CD45 (green), and E-Cadherin (purple) and nuclei staining DAPI (blue). Scale bars, 100 μm. **d** Representative FACS plots and quantification of lamina propria cells stained for CD45 and for Ccdc88b antibodies for NT ($n = 3$) and DSS-treated WT mice ($n = 4$ for each time point) at indicated time points, data are representative of one experiment. **e** Representative confocal microscopy images of tissue sections from colon of NT or DSS-treated mice at day 8, and stained for Ccdc88b (red), CD3 (green) and CD11b (green); nuclei are stained with DAPI (blue). Scale bars, 100 μm. **f** same as in **e** for Ccdc88b (red), CD4 (green) and CD8 (green). Scale bars, 100 μm

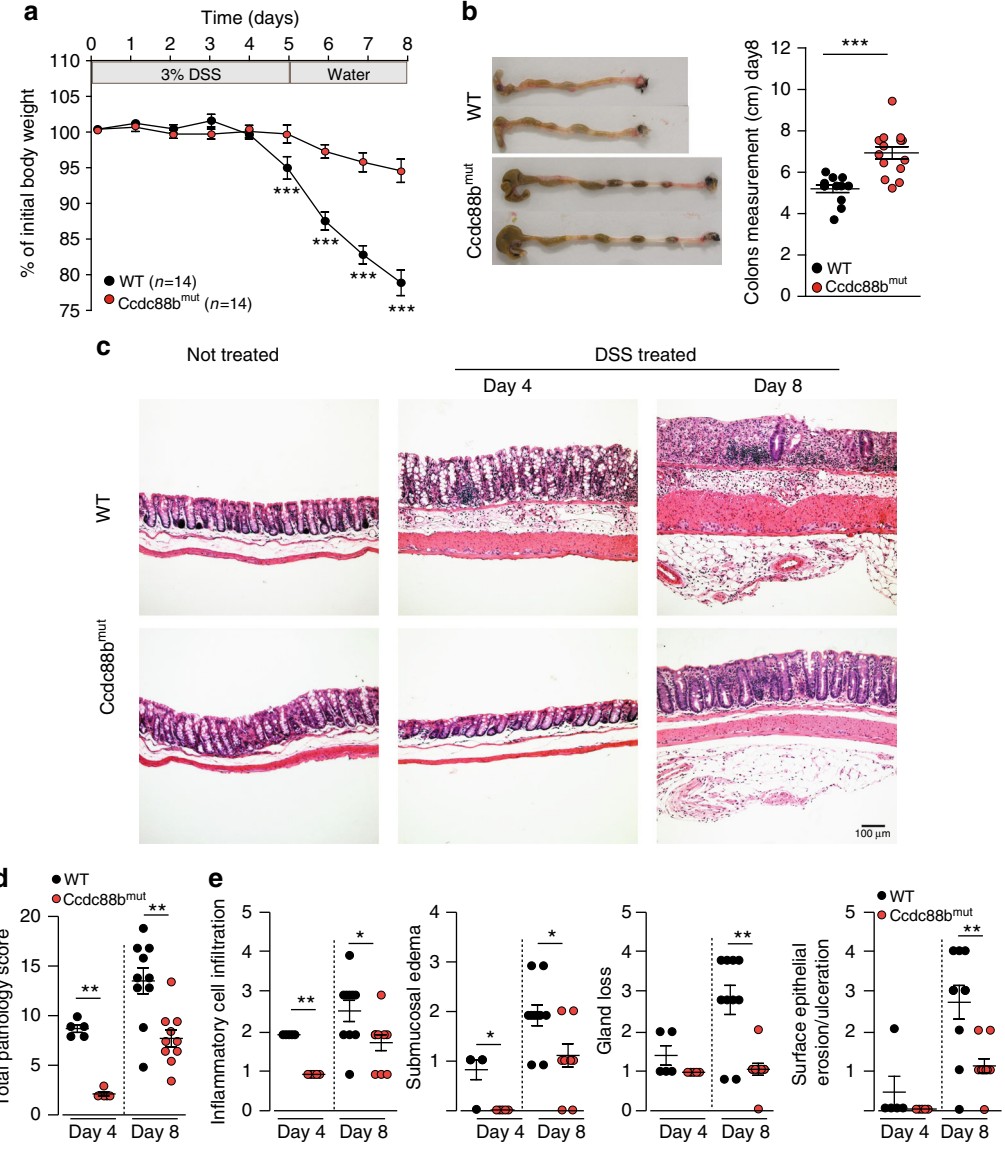

**Fig. 2** Loss of Ccdc88b protects against DSS-induced colitis. **a** WT (*n* = 14) and *Ccdc88b^mut^* mice (*n* = 14) were treated with DSS (see legend to Fig. 1) and body weight loss is expressed as percent of initial weight±SEM. ***P < 0.001 (two-tailed Student's *t*-test). **b** Representative images of colons from WT and *Ccdc88b^mut^* mice at day 8, and quantification of effect of treatment on colon length±SEM. ***P < 0.001 (two-tailed Student's *t*-test). **a**, **b** Data are pooled from 2 independent experiments. **c** Hematoxylin and eosin (original magnification ×10) staining of colon sections from untreated and DSS-treated WT and *Ccdc88b^mut^* mice at the indicated times. Scale bars, 100 μm. **d** Total pathology score (0–24) from WT and *Ccdc88b^mut^* mice at day 4 after DSS treatment (*n* = 5) and day 8 (*n* = 10) ± SEM. **p < 0.01 (Mann–Whitney test). **e** Histology scores from WT (*n* = 5 for day 4 and *n* = 10 for day 8) and *Ccdc88b^mut^* (*n* = 3 for day 4 and *n* = 10 for day 8) mice at indicated days evaluating inflammatory cell infiltration, submucosal edema, gland loss and surface epithelial erosion/ulceration and (Supplementary Table 1) for scoring details (mean±SEM; *P < 0.05, **P < 0.01, and ***P < 0.001; Mann–Whitney test)

patients and in mouse models of IBD[17–19]. We evaluated mRNA expression for different pro-inflammatory cytokines and chemokines in colons of DSS-treated mice at days 4 and 8. We found reduced expression of macrophage and neutrophil chemoattractants MCP1 and KC and of pro-inflammatory cytokines IL1β, TNF and IL6 mRNAs in colons from *Ccdc88b^mut^* mice compared to WT at day 4 after DSS treatment (Fig. 3a). Similar results were observed at day 8, with MCP1, TNF and RANTES being significantly lower in *Ccdc88b^mut^* mutants. The serum levels of MCP1 and KC, were also significantly decreased in *Ccdc88b^mut^* mice compared to WT at day 4, with this difference persisting at day 8 for KC (Fig. 3b). Compared to WT mice, the epithelium of treated *Ccdc88b^mut^* mice remained relatively intact with increased cell proliferation and greater number of mucin-containing

epithelial goblet cells assesed by Ki-67 and Alcian blue (mucus) staining, respectively (Fig. 3c, d). Hence, *Ccdc88b^mut^* mice are protected against DSS-induced colitis, exhibiting less severe pathology and reduced inflammation.

Furthermore, we crossed *Ccdc88b^mut^* mice to *Rag1^−/−^* mice and subjected these mice to an acute DSS treatment. We found that *Rag1^−/−^* mice lacking functional Ccdc88b recovered from DSS-induced colitis (increased body weight, lower colonic damage, better total pathology scores) faster and more fully when compared to control *Rag1^−/−^* mice (Supplementary Fig. 2). These results suggest that the absence of Ccdc88b in innate immune cells is sufficient to dampen epithelial barrier dammage, decrease inflammation and protect against experimental colitis.

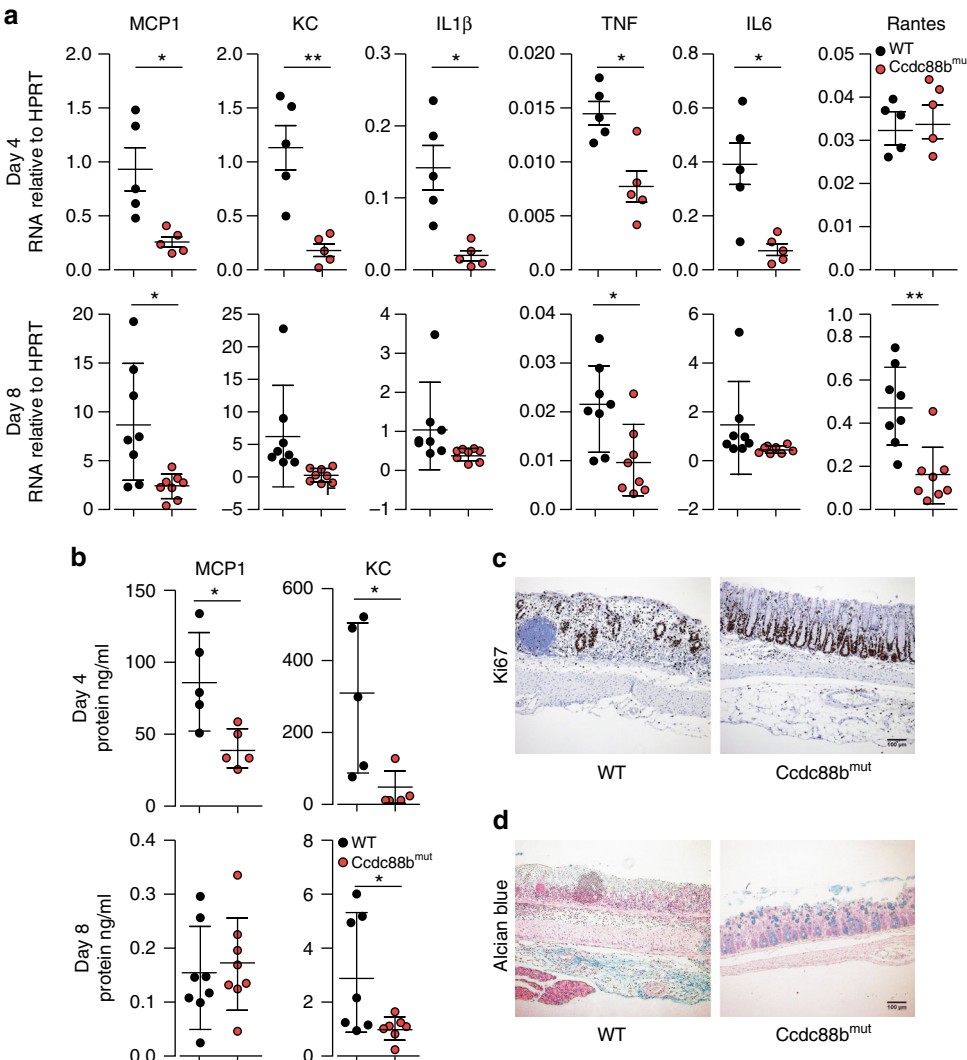

**Fig. 3** Loss of *Ccdc88b* on inflammatory response to DSS-induced colitis. WT and *Ccdc88b*^mut mice were treated with 3% DSS for 5 days followed by 3 days of water. **a** Relative expression levels of indicated cytokine and chemokine mRNAs extracted from WT and *Ccdc88b*^mut colon mice at day 4 ($n = 5$) and day 8 ($n = 10$) following DSS treatment (mean±SEM; *$P < 0.05$, **$P < 0.01$; two-tailed Student's *t*-test). **b** Proinflammatory cytokine MCP1 and KC were quantified in serum by ELISA, and results are presented in ng/ml (mean ± SEM; ***$P < 0.001$; two-tailed Student's *t*-test). **a**, **b** Data are pooled from 2 independent experiments. **c** Representative image ($n = 4$) of crypt regeneration visualized by Ki-67 staining (hematoxylin counterstaining), Scale bars, 100 µm. **d** Representative image ($n = 4$) of epithelial integrity by mucin staining (Alcian blue) of colons from DSS-treated mice at day 8. Scale bars, 100 µm

**Ccdc88b is required for T cell-induced colitis**. We have previously shown that the Ccdc88b protein is expressed in CD4+ and CD8+ T cells, and is required for T cell maturation and activation[5]. We examined the effect of *Ccdc88b* mutation on the capacity of naïve CD4+ T cell (CD4+CD45RB^hi cells, (Supplementary Fig. 3)) to induce colitis upon adaptive transfer into lymphopenic mice[20]. As previously reported[20], *Rag1*^−/− mice receiving WT CD4+CD45RB^hi cells developed symptoms of colitis including significant weight loss by week 6 post transfer (Fig. 4a). In contrast, *Ccdc88b*^mut CD4+ T cells in *Rag1*^−/−mice failed to induce similar outcomes including weight loss and colon shortening (Fig. 4a, b). Histopathological assessment (Fig. 4c, d) of both groups of *Rag1*^−/− mice receiving either *Ccdc88b*^mut or WT CD4+ T cells showed the former to display significantly less thickening of the mucosa, lower number and extent of cellular infiltration, number of abscesses and amount of glandular loss (Fig. 4d). The overall pathology score was significantly lower in *Rag1*^−/− mice receiving *Ccdc88b*^mut vs. WT CD4+ T cells, demonstrating less severe colitis in the former group (Fig. 4e). Finally, the attenuated inflammation in *Rag1*^−/−mice receiving

*Ccdc88b*^mut CD4+ T cells was associated with lower staining of T cells than observed in mice receiving WT CD4+ T cells (Fig. 4f). Our results indicate that Ccdc88b is required for the induction of gut inflammation in the T cell transfer model of colitis.

**CCDC88B expression by CD14+ cells correlates with IBD risk**. Human *CCDC88B* maps to an IBD susceptibility locus on chromosome 11 (11q13)[1]. Analysis of published genetic association data in CD patients[1] revealed significant disease association with SNPs from a 600 kb region overlapping *CCDC88B* association, with the top marker (rs641168; $p = 2.1 \times 10^{-6}$ using a logistic regression analysis) located 38.5Kb downstream *CCDC88B* transcription start site (Fig. 5a). Meta-analysis (combining GWAS and Immunochips datasets) showed a maximum association $p$ value of $4.22 \times 10^{-9}$ (top marker rs559928) at the *CCDC88B* locus[1]. Parallel analysis of *CCDC88B* mRNA expression in individual cell populations (CD4+, CD8+, CD14+, CD15+, CD19+) from PBMCs of 320 healthy individuals showed robust but variable *CCDC88B* mRNA expression in CD14+ and CD15+ cells

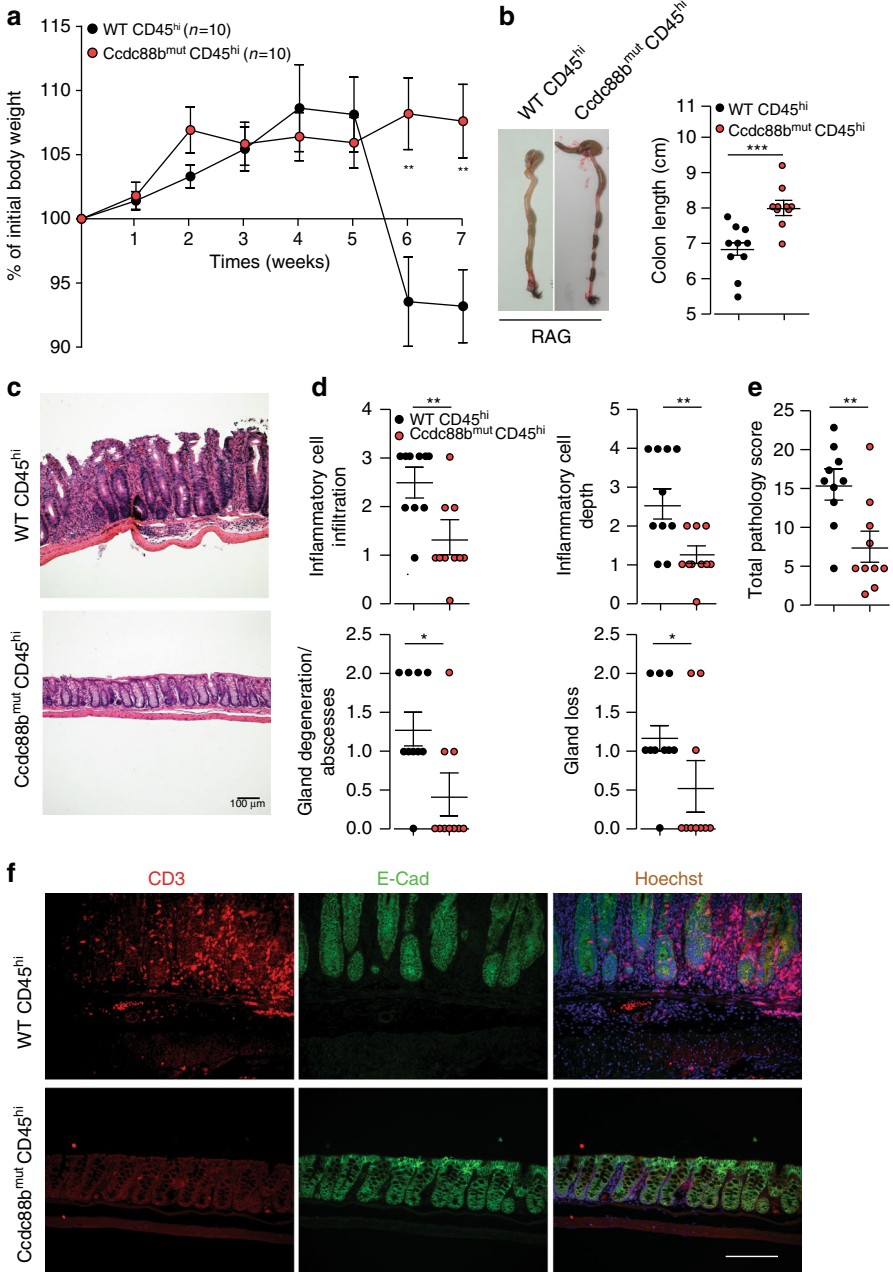

**Fig. 4** T cell–driven intestinal inflammation is attenuated in the absence of Ccdc88b. CD4+ CD25−CD45RB[Hi] T cells ($4 \times 10^5$) from WT (WT CD45[hi]) or from *Ccdc88b[mut]* (Ccdc88b[mut] CD45[hi]) mice were transferred into *Rag1[−/−]* and mice were monitored for appearance of clinical symptoms. **a** Weight of mice expressed as percentage of initial weight (mean±SEM; **$P < 0.01$, two-tailed Student's $t$-test) ($n = 10$ mice per group, data are pooled from 2 independent experiments). **b** Representative images of colons from mice 7 weeks post-T cells transfer and measurement of colon length (mean±SEM. ***$P < 0.001$, two-tailed Student's $t$-test). **c** Hematoxylin and eosin (original magnification ×10) staining of colon sections at week 7 after transfer. Scale bars100 μm. **d** Indicated colon histology scores for *Rag1[−/−]* mice having received either CD45[hi] WT or *Ccdc88b[mut]* T cells at week 7, ($n = 10$ per group; mean±SEM. *$P < 0.05$, **$P < 0.01$ Mann–Whitney test). **e** Total pathology score (0–36) (mean±SEM. **$P < 0.01$, Mann–Whitney test) and (Supplementary Table 1) for scoring details. (**a–e**) Data are pooled from 2 independent experiments. **f** Representative immunofluorescence images ($n = 3$) of CD3+T cells (red), E-cadherin (green) and nuclei with Hoechst (blue) of indicated *Rag1[−/−]* mice at week 7 post-transfer. Scale bars, 50 μm

(Supplementary Fig. 4a). eQTL analysis can be used to identify relevant variant:gene pairing among potential combinations of SNPs mapped by GWAS and the adjacent genes[21]. Because genotype information was available for these individuals, we analyzed the possible effect of 11q13 SNPs near *CCDC88B* on expression of the gene in these PBMC cell populations. We noted an effect of several SNPs in this region on *CCDC88B* mRNA expression in CD14+ cells, with the top marker (rs146881600; *p* = 0.0016) mapping at 6Kb from rs641168 and 32 Kb from

*CCDC88B* (Fig. 5b). If the genetic associations observed for this chromosomal region with CD on the one hand and *CCDC88B* expression on the other hand are causally related, one predicts that the corresponding association patterns (i.e. the log(1/*p*) values for all SNPs in the region for CD and *CCDC88B* expression, respectively) may be similar. Thus, we investigated a possible correlation between the strength of the cis-acting eQTLs effect on *CCDC88B* expression in CD14+ cells, and strength of the association of the same SNPs with disease in our cohort of CD

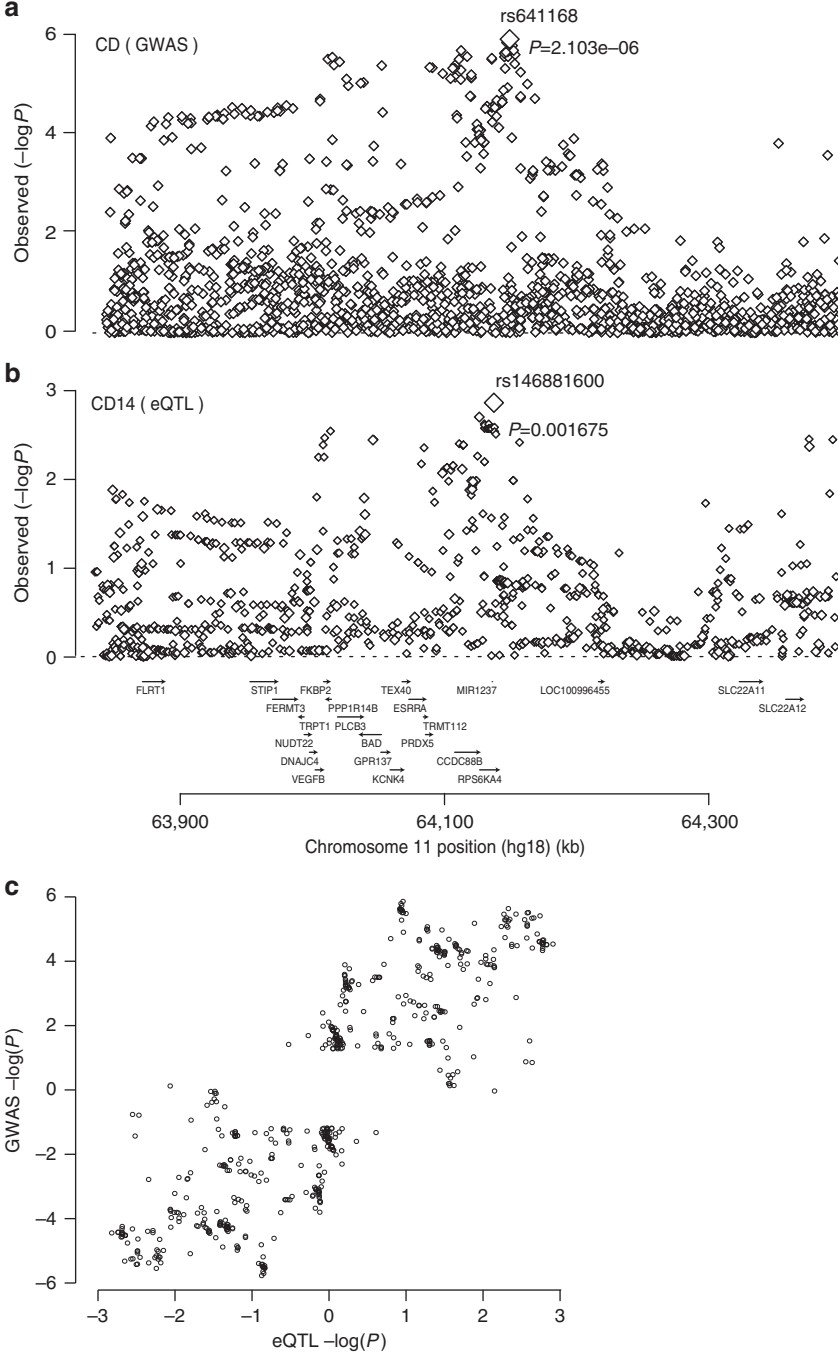

**Fig. 5** CD risk is positively associated with expression of CCDC88B in CD14[+] cells. **a** Genetic association between CD and SNPs in a 600Kb window on 11q13. Log(1/p) values are given for all SNPs positioned on the human hg18 build[1]. The association signal was maximal for rs641168 (P = 2.1 × 10[-6], using a logistic regression analysis). Gene names and positions are shown under the graph. **b** eQTL association pattern for CCDC88B expression in CD14[+] cells for the same chromosome region as in **a**. The association signal was maximal for rs146881600 (P = 0.0017, using a logistic regression analysis). **c** Correlation between the disease and eQTL association patterns. Two data points are shown for each "informative" SNP (i.e., with data in both analyses and log(1/p) value ≥1.2 with either disease, eQTL or both). One corresponds to log(1/p) values (disease and eQTL) multiplied by the signs of the effects of allele 1, the other with the log(1/p) values multiplied by the signs of the effects of allele 2. The observed correlate was shown to be significant (P = 0.01, by a logistic regression analysis) using the approach described in the Methods section

patients (Fig. 5c) (Equation 1, Methods). This analysis identifies a significant correlation (accounting for LD in the region) between eQTLs effects on *CCDC88B* expression and association (in GWAS dataset) with CD disease risk (p > 0.01), with increased *CCDC88B* expression associated with increased risk (Fig. 5c). A similar analysis performed for CD4[+], CD8[+], CD15[+] and CD19[+] cell subsets failed to identify such a correlation between eQTL effects on *CCDC88B* mRNA expression in these cell subsets and association with CD risk (in GWAS datasets) (Supplementary Fig. 4b). These results strongly suggest that differences in the level of expression of *CCDC88B* expression (eQTL) in CD14[+] cells are associated with variable risk of IBD.

**CCDC88B is increased in colons from IBD patients**. We further evaluated CCDC88B mRNA and protein levels in colon surgical specimens obtained from UC patients (UC; $n = 46$) and CD patients (CD; $n = 11$), and from normal controls, consisting of non-inflamed areas of the colon of patients with colorectal cancer or diverticulitis ($n = 73$). We found a higher expression of

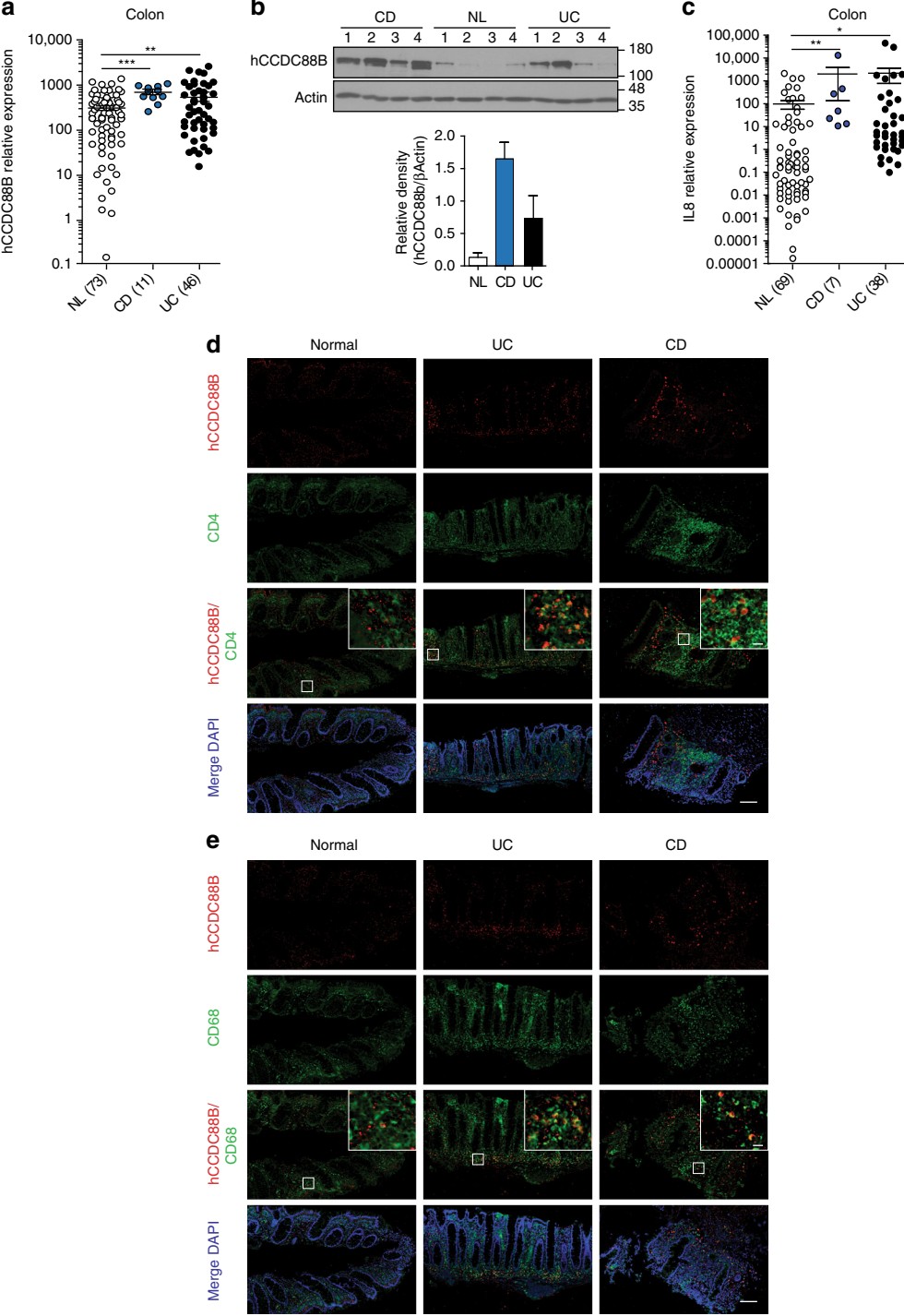

**Fig. 6** *CCDC88B* expression in inflamed colons from patients with IBD. **a** *CCDC88B* mRNA expression in colon tissue samples extracted from surgical specimens of patients with Crohn's Disease (CD, $n = 11$), Ulcerative colitis (UC, $n = 46$) or normal control (NL, $n = 73$). Data represent expression of *CCDC88B* relative to the ribosomal protein L32 (mean±SEM; **$P < 0.01$, ***$P < 0.001$; two-tailed Student's $t$-test). **b** Immunoblot analysis of CCDC88B and β-ACTIN performed on colonic tissue extracts obtained from surgical specimens of patients with CD ($n = 4$), UC ($n = 4$) or NL controls ($n = 4$). Each lane denotes an individual patient sample. Densitometric quantification of CCDC88B and β-actin immunoblots. Bar graphs represent normalized relative density of each lane ($n = 4$ patient samples). **c** *IL-8* mRNA expression in colon tissues samples from CD ($n = 69$), UC ($n = 7$) and NL ($n = 38$). Data represent expression of *IL8* relative to the ribosomal protein L32 (mean±SEM; **$P < 0.01$, ***$P < 0.001$; two-tailed Student's $t$-test). **d, e** Immunofluorescence staining of CCDC88B, CD4 and CD68 in colon tissue sections from patients with CD, UC, or NL controls (CCDC88B, red; nuclei blue staining with Hoechst). Isotype control staining for each section is shown. Scale bars, 100 μm and 10 μm

*CCDC88B* mRNA in the colon of patients with IBD in both CD and UC in comparison with non-inflamed mucosa (Fig. 6a). By immunoblotting, using a specific antibody against human CCDC88B (Supplementary Fig. 5) in a subset of colon surgical specimens, we found CCDC88B protein significantly increased in inflamed colons of CD and UC patients compared to non-inflamed controls (Fig. 6b and Supplementary Fig. 7b). We also quantified expression of the pro-inflammatory cytokine IL-8 in a subset of clinical specimens consisting of 69 controls, 7 CD and 38 UC samples; we deetcted increased IL-8 expression in specimens from CD and UC patients compared to NL controls (Fig. 6c).

We used immunohistochemistry to examine the site of CCDC88B protein expression in inflammaed mucosa from IBD patients. First, to confirm the inflamed nature of the tissue samples used in our experiments, blind histological scoring was performed on H&E sections. Apparent chronic inflammation and active inflammation were observed in the colon mucosa of UC and CD patients (Supplementary Fig. 6a), and was paralleled with increased presence of CD3[+] and CD8[+] cells (Supplementary Fig. 6b). CCDC88B expression in inflamed mucosa was limited to non-epithelial cells in the lamina propria, and colocalised with CD4[+] and CD68[+] infiltrating immune cells (Fig. 6d, e). These CCDC88B[+] cells were found in greater numbers in the mucosa from patients with CD and UC compared to non-inflamed controls (Fig. 6d, e). Hence, our data show that human CCDC88B is highly expressed in colon lamina propria cells of inflamed mucosa from UC and CD patients, with its expression limited to infiltrating lymphoid and myeloid cells.

## Discussion

We have previously shown that a mutation in *Ccdc88b* protects mice against lethal neuroinflammation in models of microbial disease[5]. In these studies, the loss of *Ccdc88b* did not affect the number of lymphoid and myeloid cells, but it impacted the maturation, and activation of these cells including production of pro-inflammatory cytokines[5]. Here we show that a mutation in *Ccdc88b* protects mice against experimental intestinal colitis, both in the DSS-colitis and CD4 T cells transfer models. Studies of the mouse model of colitis have clearly shown that reduced inflammation in situ decreases pathogenesis[22]. In these models, it has been reported that secretion of of IL-1β, IL-6, IL-9, and TNF drives pathogenesis[22–25]. In the absence of Ccdc88b, we found that the protection is associated with reduced recruitment of hematopoietic cells to the site of inflammation, concomitant to reduced local and systemic production of pro-inflammatory cytokines. In the CD4 T cell transfer model[20], absence of Ccdc88b expression in CD45RB[hi] CD4 T cells is associated with protection against intestinal inflammation.

Leukocytes trafficking and recruitment into the inflamed intestine is fundamental to the development of IBD, including functional interaction between the α4β7integrin expressing T cells and MAdCAM-1[+] gut endothelial cells[26–29]. How does CCDC88B may regulate immune cells function during intestinal inflammation? An interesting clue comes from a recent report documenting physical interaction between CCDC88B (HkRP3) and the CDC42 guanine nucleotide exchange factor DOCK 8[7]. *DOCK8* mutations cause primary immunodeficiencies that clinically manifest as severe allergy, skin and lung infections in humans[8,9]. DOCK8 has been shown to be essential for the homing and function of myeloid (dendritic cells) and NK cells[8,9]. In addition, CCDC88B (HkRP3) is required for NK cells cytotoxicity and anti-tumor activity in general, and for the production and mobilization of cytotoxic granules in these cells[7]. It is tempting to speculate that CCDC88B may regulate migratory properties of myeloid and lymphoid cells during intestinal inflammation.

We also show that human CCDC88B mRNA and protein are increased in colon specimens from IBD patients, and linked to presence of CCDC88B[+] cells in these tissues. In agreement with previously published results[30], we observe that (a) *CCDC88B* is genetically linked to susceptibility to IBD, and (b) establish that the level of *CCDC88B* mRNA in normal CD14[+] but not to other cells subsets is regulated by cis-acting variants (eSNPs), (c) detect a significant correlation between eQTLs effects and disease risk, with increased *CCDC88B* expression associated with increased risk and (d) observe that CCDC88B[+]myeloid and lymphoid cell subsets infiltrate the lamina propria of CD and UC patients. The specificity of the eQTL effect on CCDC88B mRNA expression in human CD14[+] myeloid cells and its association with IBD disease risk may seem in apparent contrast to the results obtained in the RAG/lymphocyte deficient reconstitution model and that support a role of CCDC88b in T cells function in this mouse model of intestinal inflammation. We believe this points to a key role of CCDC88B in both lymphoid and myeloid cells, with the former being critical in this mouse model of inflammation, while the latter is possibly more relevant to the human disease and the criteria used to establish diagnosis. Indeed, we have previously demonstrated a role for CCDC88B in the function of both T cells and myeloid cells in other mouse models of inflammation (neuroinflammation)[5].

Our results are compatible with a model in which CCDC88B is a microtubule associated protein which physically and functionally interacts with DOCK8, a protein required for mobility and homing of lymphoid and myeloid cells. We propose that genetically regulated levels of CCDC88B in immune cells may impact the recruitment of these cells to the site of tissue injury, causing variable inflammatory response in situ, and variable disease pathogenesis. This model is consistent with the pleiotropic effect of *Ccdc88b* mutation in different experimental models of inflammation in mice, and the association of the 11q13 locus with several inflammatory diseases in humans.

## Methods

**Ethics statement**. All experiments with live mice conducted in this study were performed in accordance and compliance with the guidelines and regulations of the Canadian Council on Animal Care (CCAC). All protocols were approved by the Animal Care Committee of McGill University (protocol number 5287; Principal Investigator: P. Gros). Mice were anesthetized, and euthanized by carbon dioxide inhalation, and every effort was made to minimize animal suffering. Sample size to achieve adequate power was chosen on the basis of previous studies with similar methods[20,31,32]. Briefly, a minimum of 3 mice for control animals and between 4 to 7 mice (per experiment) were used for treated animals. We randomized mice from different cages and different time points to exclude cage or batch variation. Experiments were not performed in a blinded fashion except when specifically indicated. Experiments and data analysis was performed without exclusion criteria, and all data was retained.

For isolation of human PBMC, informed consent was obtained from all donors and approved by the governing ethics committee (Comité d'ethique hospital-facultaire universitaire de Liège; Protocol B70720097536; 2009/256; Principal Investigator: Prof. E. Louis). For clinical specimens, only disposed de-identified segments of surgical specimens from patients undergoing bowel resection as part of their medical care for ulcerative colitis, Crohn's disease or colorectal cancer/diverticulitis (controls)at the Mount Sinai Medical Center had been used in this study. The obtained specimens were pathological waste and used for research purposes only. Surgical wastes were completely anonymous and patient's identifiers had never been communicated to the research team in order to strictly maintain patient's confidentiality. Once unidentified specimens arrived in the laboratory, they were given lab accession numbers, which were not linked to any of the patient identifiers directly or indirectly. Because of the above process, the institutional review board at the Mount Sinai Medical Center considered this study as being "none human subject" research which fell under the waiver of the Consent Process. As a result, consent forms were not required for this study.

**Mice**. Wild type C57BL/6 J, *Ccdc88b*^m1PGrs^ (*Ccdc88b*^mut^)[5] and *Rag1*^−/−^ (RAG) mutant male mice (on B6 background), 8 to 10 weeks of age were obtained from

the Jackson Laboratory (Bar Harbor, ME) and were housed under specific pathogen-free conditions at the animal care facility of the Goodman Cancer Research Centre, McGill University. The animal studies were conducted under protocols approved by the McGill Institutional Review Board (protocol number 5287), and following guidelines and regulations of the Canadian Council of Animal Care.

**Mouse models of intestinal colitis.** To induce colitis mice (8 weeks old, male) were fed 3% (w/v) DSS (dextran sodium sulfate; molecular mass 36–40 kDa; MP Biomedicals) in the drinking water for 5 days. This was followed by normal drinking water until day 8, at which point animals were sacrificed. Mice were co-housed at weaning until the start of the experiment. The animals were weighed daily and monitored for signs of distress. For T cell transfer-induced colitis, naive CD4$^+$ CD45RB$^{hi}$ T cells from WT or *Ccdc88b$^{mut}$* mice were enriched (CD4$^+$T Cell Isolation Kit; Miltenyi Biotec) and single-cell suspensions were stained with antibody dilutions 1:300 FITC-anti-CD4 (GK1.5), 1:250 PE-anti-CD25 (PC61.5) and 1:1500 APC–anti-CD45RB (C363.16 A), all from eBioscience[33], following purification (>99%) by cell sorting (FACSAriaII). Sex-matched *Rag1$^{-/-}$* recipient mice received $5 \times 10^5$ CD4$^+$ CD45RB$^{hi}$ T cells by intravenous (i.v.) injection. Mice were sacrificed when symptoms of clinical disease (significant weight loss) became apparent in control mice receiving WT cells, ~7 weeks after injection.

**Lamina propria cell preparation and flow cytometry analysis.** Colons were collected from control and DSS-treated mice and tissues were dissociated with Lamina Propria Dissociation Kit, (Miltenyi Biotec), according to the manufacturer instructions. Single-cell suspensions were prepared and cells were first stained with vital dye Zombie Aqua™ dye (at 1/400 dilution) (Biolegend) then surface stained with the following fluorescently-labeled antibodies: 1:300 PerCP-Cy5.5-Ly6G (1A8), 1:200 PE-Dazzle 594-CD3 (17A2), 1:100 BV421-Ly6C (HK1.4) all from Biolengend, 1:300 PE-Cy7-CD4 (GK1.5), 1:200 APC-NK1.1 (PK136), 1:200 Alexa 700-CD8 (53–6.7), 1:300 Evolve 605-CD11b (M1/70) and 1:300 APC-Efluor-780-CD45 (30-F1) all from eBiosicense. After permeabilization, cells were stained with rabbit anti-mouse Ccdc88B and anti-rabbit Alexa 488. Stained samples were analyzed on a Fortessa flow cytometer (BD Bioscience) and the results analyzed using FlowJo software (Tree Star Inc.).

**Anti-human CCDC88B antibody production and purification.** Polyclonal rabbit anti-human CCDC88B antiserum was prepared as we have previously described[5,34]. A recombinant protein consisting of glutathione-S-transferase (GST) fused in-frame to a CCDC88B segment corresponding to human amino acid positions 650 to 769, was expressed in *E. coli* BL21, followed by affinity purification with glutathione-agarose beads (GE Healthcare). Antisera were raised in New Zealand white rabbits using purified protein (0.5 mg per rabbit per injection) emulsified in Freund's incomplete adjuvant. Affinity purification of the anti-CCDC88B antibody[35] was using a recombinant CCDC88B protein (positions 650 to 769) comprising a poly-histidine tail (His)6 fused in-frame at its N terminus, and purified by chromatography onto Ni-NTA agarose (Qiagen). The immobilized protein was used to capture the anti-CCDC88B fraction of the hyperimmune serum, which was then released by washing with imidazole-containing buffer[34]. The specificity of the anti-CCDC88B antibody was tested by western blotting total cell extracts from HEK293T control cells and HEK293T cells (ATCC, CRL-11268) stably expressing a full-length human *CCDC88B*.

**Immunoblotting.** Tissue homogenates were prepared in lysis buffer solution (150 mM NaCl, 20 mM Tris (pH 7.4), 10 mM EDTA, 1.5 mM EGTA and 1% Nonidet P-40) supplemented with a cocktail of protease inhibitors; 50 mM NaF, 10 mM of $Na_4P_2O_7$;1 mM $Na_3VO_4$ in addition to 1X of protease inhibitor cocktail tablet (Roche) using a Polytron Homogenizer (Kinematica AG). Samples were clarified, denatured with SDS buffer, and boiled for 5 min. Proteins were separated by SDS-PAGE and transferred on Immobilon®-P PVDF Membrane (Millipore). The membranes were immunoblotted with primary rabbit anti-mouse or rabbit anti-human CCDC88B antibodies followed by detection with anti-rabbit antibodies.

**Immunohistochemistry.** For Ki-67 staining, formalin-fixed paraffin-embedded tissue sections were de-waxed and rehydrated, incubated in Diva Decloaker antigen retrieval solution (Biocare) and boiled for 20 min in a pressure cooker. Peroxidase activity was blocked using the Enzyme Block for 15 min (DAKO). Slides were stained with 1:400 anti-Ki-67 (D3B5; Cell signaling) for 1 h followed by another hour of incubation with 1:500 of goat anti-rabbit (Jackson laboratories). Finally, slides were incubated with DAKO DAB substrate (DAKO) for 10 min and counterstained with hematoxilin, then mounted. For Alcian Blue staining (1% solution, Sigma), slides were stained at room temperature for 30 min and then washed. Cell nuclei were stained with Kernchtrot Nuclear Fast Red stain (1 min). Slides were washed with distilled water, dehydrated with anhydrous alcohol and mounted.

**Immunofluorescence.** Tissues for cryopreservation were fixed in 4% paraformaldehyde (PFA) for 6 h, incubated in 20% sucrose overnight, and frozen in

Optimal Cutting Temperature (OCT) medium (VWR International). For immunofluorescence, frozen sections (7 µm) were washed in phosphate-buffered saline (PBS), incubated in blocking buffer (10% BSA, 0.4% Triton-X-100 in PBS) for 1 h at room temperature, and then incubated with primary antibody in primary antibody solution (1% BSA, 0.4% Triton-X-100) in phosphate-buffered saline (PBS) for 1 h at room temperature or overnight at 4 °C. The primary antibodies and dilutions used were as follows: 1:200 rabbit antiCcdc88b[5], 1:100 rat anti-CD3ε-FITC (145-2C11, BioLegend), 1:100 rat anti-CD11b-APC (M1/70, eBioScience), 1:100 rat anti-CD4-FITC (GK1.5, BioLegend) 1:100 rat anti-CD8-FITC (53–6.7), BioLegend) 1:100 rat anti-E-Cadherin-660 (DECMA-1, eBioScience), 1:100 rat α-CD45-PE (30-F11, eBioScience). Bound antibodies were detected using a 1:1000 dilution of rabbit anti-Cy3, or AlexaFluor488. Immunostained sections were counterstained with 4′,6-diamidino-2-phenylindole (DAPI) at a 1:3000 or Hoechst at 1:5000 to visualize the nuclei. Images were acquired on a Zeiss LSM710 Meta Laser Scanning Confocal microscope or Axioscope 2 plus, processed with ImageJ, and compiled with Adobe Illustrator. Paraffin-embedded sections of NL, CD and UC specimens were heat treated in citrate buffer (pH 6) for 10 min at 121 °C for antigen retrieval. The section were blocked with 10% normal human serum in PBS for 1 h at room temperature. Slides were stained with a combination of 1:100 rabbit anti-hCCDC88B and 1:150 mouse anti-CD68 (KP1, Abcam) or mouse 1:100 anti-CD4 (N1UG0, Affimetrix) to assess colocalization or 1:150 rabbit anti-CD3 (SP7, Abcam) and rabbit anti-CD8 (NBP2-29475, Novus Biologicals). Bound antibodies were revealed using a dilution of 1:500 of mouse anti-AlxaFluor 488 or rabbit anti-AlxaFluor 647 and counterstained with DAPI. Images were aquiered using Olympus VS110 systems and analyzed and processed with Olyvia software (Olympus).

**Gene expression and cytokine quantitation in colon tissues.** RNA was purified using a FastPrep 24 homogenizer (MP Biomedicals) with lysing matrix D beads (MP Biomedicals) and RNAeasy kits (QIAGEN). RNA purity and quantification was determined using a Nanodrop spectrophotometer (Nanodrop Technologies). cDNA synthesis was performed using a MMLV reverse transcriptase with a cocktail of Oligo dT and random hexamers (Invitrogen). Human and mouse *Ccdc88b* mRNA expression were quantified using the following primer sets, m*Ccdc88b* (5′-CCGGGAGCTTCGAGGGCCAAC-3′ and 5′-CCTATCTGGCAAG CGGGGC-3′) and h-*CCDC88B* (5′-GCGTGAGGGGTCCAGGC-3′ and 5′-CTCC TTGCCCCGGCACCAC-3′). Cytokines and chemokines were quantified by RT-qPCR using Perfecta SYBR Green PCR kit and the specific primers sets m*MCP*-1/ *CCL2* (5′-AGGTGTCCCAAAGAAGCTGTA-3′ and 5′-TCTGGACCCATTCC TTCTTG-3′), *KC*/ *CXCL1* (5′-CACCTCAAGAACATCCAGAGC-3′ and 5′-CTTG AGTGTGGCTATGACTTCG-3′), m*IL1-β* (5′-CGGCACACCCACCCTG-3′, and 5′- AAACCGCTTTTCCATCTTCTTCT-3′) m*TNF*- (5′-TCTCAGCCTCTTCTCA TTCC-3′ and 5′-AGAACTGATGAGAGGGAGGC-3′), m*IL-6* (5′-GAAGTAGGG AAGGCCGTGG-3′ and 5′-GAAGTAGGGAAGGCCGTGG-3′), m*Rantes*/ *CCL5* (5′-GCAAGTGCTCCAATCTTGCA-3′ and 5′-CTTCTCTGGGTTGGCACACA-3′), h*IL-8* (5′-CTGGCCGTGGCTCTCTTG-3′) and (5′CTTGGCAAAACTGCAC CTTCA-3′). Gene expression was normalized to h*L32* expression (5′-TGTCCTGA ATGTGGTCACCTGA-3′ and 5′-CTGCAGTCTCCTTGCACACCT-3′) and m*Hprt* (5′-TCAGTCAACGGGGGACATAAA-3′ and 5′-GGGGCTGTACTGCTT AACCAG-3′) and relative expressions calculated using the ΔΔCT method. Serum was prepared from blood of DSS-treated animals. MCP-1/CCL2 and KC/CXCL1 chemokine were quantified by ELISA (R&D Systems) according to the manufacturer's protocol.

**Assessment of intestinal inflammation.** At experimental endpoints, the entire colons were excised, and their length measured. Colons were fixed overnight in 10% PFA, embedded in paraffin, and 4 µm sections were stained with hematoxylin & eosin. Histology was evaluated blindly by a pathologist to avoid bias. For DSS-induced colitis, histology was scored from 0 to 4 for a combination of inflammatory cell infiltration, surface epithelial erosion / ulceration, and gland loss for a total score of 24. For the transfer of naive CD4$^+$T cells, histology was scored from 0 to 4 for the following criteria; inflammatory cell infiltration, inflammatory cell depth, submucosal edema, increased mucosal thickening, surface epithelial degeneration, gland epithelial apoptosis, gland epithelial degeneration/abscesses, gland goblet/ enterocyte ratio decrease, submucosal edema and gland loss for a total score of 36 (detailed pathology scoring in Supplementary Table 1).

**Clinical specimens.** Disposed anonymous surgical specimens from patients undergoing bowel resection for Crohn's disease, ulcerative colitis or colorectal cancer/diverticulitis (controls) at the Mount Sinai Medical Center were acquired as previously published[36,37]. Both inflamed CD or UC, and non-inflamed controls were used. RNA was extracted from frozen colonic specimens corresponding to active Crohn's Disease (CD) and Ulcerative Colitis (UC) mucosal tissue along with adjacent normal tissue (Adj, >10 cm from the lesion) from the same patients.

**Expression-QTL analyses and data.** EDTA-treated blood from 320 individuals were layered on Ficoll-Paque PLUS (GE Healthcare) for isolation of peripheral blood mononuclear cells by density gradient centrifugation. CD4$^+$, CD8$^+$, CD19$^+$, CD14$^+$, and CD15$^+$ cells were isolated using MicroBead Technology (Miltenyi

Biotec). Total RNA was isolated using the AllPrep Micro Kit on a QIAcube robot (Qiagen). Whole genome expression data were generated for each of the cell/tissue types on Illumina Human HT-12 Expression BeadChips. 8,537 out of 47,231 probes mapping to multiple locations or encompassing known SNPs with MAF ≥0.05 were eliminated from further analyses. Raw fluorescent intensities were variance stabilized[38] and quantile normalized[39] using lumi R package[40]. Normalized expression data were corrected for Sentrix ID (random effect), age, sex, smoking status, and number of probes exceeding the detection limit (fixed effects and covariates, if significant ($p \leq 0.05$) using a mixed model implemented with lme4 R package. eQTL analyses were performed using PLINK.

**Genetic association between CD and CCDC88B expression**. The correlation between the genetic association with gene expression level and disease assumes that; (a) genetic variant(s) in a given chromosome region affect the levels of expression of gene(s) in that region and that this affects predisposition to a given disease (CD in this study), one can predict that the corresponding association patterns (i.e. the set of $\log(1/p)$ values of association for the SNPs in the region with CD and gene expression, respectively) will be positively correlated. Such positive correlation can correspond to two distinct scenarios: increased gene expression may either be associated with increased or decreased disease risk. This information can be captured by calculating the correlation between $\log(1/p)$ values multiplied by the sign of the corresponding genetic effects (β). To avoid that the ensuing metric would be sensitive to the choice of the reference allele, we duplicate each data point in the correlation analysis, respectively considering the two alleles as reference in the association analysis. In summary, we generate a correlation metric, D(isease)E(xpression)C(orrelation), corresponding to:

$$DEC = Pen \times SC(x, y) \tag{1}$$

Pen is a penalty function that measures the strength of the positive correlation between the $\log(1/p)$ values for the disease and eQTL associations respectively. Pen is defined as:

$$Pen = 1 / \left( 1 + e^{-k1 \times (\sigma(x,y) - k2)} \right) \tag{2}$$

$\sigma(x,y)$ corresponds to Spearman's rank correlation between the $\log(1/p)$ values of associations with disease and gene expression for the SNPs in the region, and k1 and k2 to constants that determine the severity of the penalty, and which were empirically set at 30 and 0.3, respectively, in this study. We restrict the analysis to "informative" SNPs that have a $\log(1/p)$ value >1.2 for either the disease, or the eQTL or both. SC(x,y) stands for "signed correlation". It corresponds to Spearman's rank correlation between two pairs of sign_β*$\log(1/p)$ values for each SNP in the studied region. The first pair corresponds to the $\log(1/p)$ values multiplied by the sign of the effect of the association on disease and gene expression, respectively, assuming allele 1 as reference. The second pair is identical but assumes allele 2 as reference. We restrict the analysis to "informative" SNPs that have a $\log(1/p)$ value >1.2 for either the disease, or the eQTL or both.

To measure the statistical significance of the observed DEC, yet accounting for the LD structure of the studied region, we generated as many eQTL as there are SNPs in the studied region in silico. The in silico eQTL are simulated such that they explain the same variance as the true eQTL. For each simulated eQTL with compute a corresponding "simulated DEC". The statistical significance of the actual DEC is then measured as the number of simulated DEC that are as large or larger than the true DEC. The $\log(1/p)$ values for CD were directly obtained from[1]. The $\log(1/p)$ values for the eQTL were obtained from the dataset described above.

**Statistical analysis**. Results are presented as mean±SEM. Prism5 (GraphPad) software was used for all statistical tests. Statistical significance was determined by two-tailed Student's $t$-test for comparison of two groups and Mann–Whitney test for intestinal scoring[41], unless otherwise indicated. Differences were considered statistically significant when $P \leq 0.05$. $P$-values are indicated by *$P < 0.05$, **$P < 0.01$, ***$P < 0.001$.

**Data availability**. The data that support the findings of this study are available within the article, its Supplementary Information files and from the corresponding author upon reasonable request.

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

## Acknowledgements

This work was supported by research grants to PG from the Canadian Institutes of Health Research; grants from WELBIO (Grant CAUSIBD) and from BELSPO (PAI BeMGI) for M.G., E.T., Y.M., J.D. and E.D. Work in the Shoukry lab was supported by pilot project funding from Fonds de Recherche du Québec–Santé (FRQS) AIDS and Infectious Disease Network (Réseau SIDA-MI) and the Canadian Network on Hepatitis C (CanHepC). MFM received fellowships from the Université de Montréal, Bourse Gabriel Marquis and the FRQS. NHS is supported by a Chercheur Boursier salary award from the FRQS. G.Y.'s work at Mount Sinai was supported by The Leona M. and Harry B. Helmsley Charitable Trust. We thank Dr. Petronela Ancuta (Universite de Montreal) for sharing reagents. We thank Maryse Dagenais, Alexandre Morizot, and Claudia Champagne (McGill University) for technical help. The authors are indebted to Genevieve Perreault for expert technical assistance. S.D. is currently is employed by Sobi, Inc.; the article in no way represents the work product, views or opinions of Sobi, Inc. G.Y. is currently employed by The Leona M. and Harry B. Helmsley Charitable Trust; the article in no way represents the work, views or opinions of Helmsley.

## Author contributions

N.F., N.M. and G.Y., performed western blots, N.F., V.L, M.F.M. and N.H.S., performed IHC experiments, N.F., J.-F.O. and I.R., performed DSS and Flow cytometry experiments, N.F., T.J., A.M., D.B. and G.Y., performed RT-qPCR experiments, R.C. performed histological scoring, N.F. performed CD4+ T cell transfer experiments, M.K., J.D., E.L., E.T., Y.M. and M.G., performed eQTL analysis and genetic studies, S.D. G.Y., provided tissue samples, N.F., M.G., G.Y. and P.G. prepared the manuscript, N.F. and P.G. designed the experiments, conceived and directed the research.

## Additional information

**Competing financial interests::** The authors declare no competing financial interests.

