## [Peer Review File · Nature Communications]

Editorial Note: Parts of this peer review file have been redacted to maintain the confidentiality of unpublished data.

Reviewers' comments:

Reviewer #1 (Remarks to the Author):

Interesting study with both mice and human data making these findings potentially clinically relevant

1. Please use another model of experimental colitis (TNBS or oxazolone)
2. Methods: What is the reference of the primary and secondary antibodies used for detection of CCDC88B by WB?
3. Figure 1. Please perform some double immunostaining to investigate which type of T cells are involved here (CD4+ or CD8+)
4. Regarding KO mice, please show the expression levels of CCDC88B in mice colon
5. Figure 6: Please investigate which type of cells are involved here similar to Figure 1
6. Human data: please show the correlation between inflammatory markers and expression levels of CCDC88B
7. Human data: colorectal rectal cancer or diverticulitis is not a good control. Please non-inflamed areas (macroscopically and histologically) from colon of IBD patients
8. Figure 4: Please look at expression levels of pro- and anti-inflammatory cytokines within colon
9. Methods: please give further details on the histological score used for animal experiments

Reviewer #2 (Remarks to the Author):

This study examines the effects of CCDC88b on two models of intestinal inflammation, namely the DSS and T cell transfer model. They demonstrate that in both models, mutant CCDC88b protects against colitis. In the DSS model, they demonstrate an induction of CCDC88b expression. This group's interest in this molecule stems from their prior work (Kennedy et al., JExpMed 2014) using an ENU screen to identify factors that protect against cerebral malaria. In that prior work, they demonstrated that CCDC88b mutants demonstrate impaired T cell maturation/differentiation. CCDC88b is expressed in both lymphocytic and myeloid cell populations. They also show data that from PBMCs from 320 individuals that there is an eQTL in CD14+ peripheral blood monocytes that correlates with the maximally disease-associated SNPs. Finally, they show expression data from IBD intestine demonstrating induction of CCDC88B in non-epithelial cells of the colon from both CD and UC.

The strengths of this study include the relative novelty of this understudied protein and the compelling role that CCDC88b plays in the T cell transfer model of colitis. Given the paucity of functional analyses of IBD-associated loci, this study represents a much needed functional interrogation of disease-associated loci in IBD.

The major weaknesses/interpretative limitations of the study include

1) The eQTL analysis of peripheral blood (Figure 5). The analyses provided are limited due to the fact that the authors only present the results from CD14+ cells, despite having generated data from CD4+, CD8+, CD19+ and CD15+ cells. Even if negative, and even if the authors don't see compelling inter-individual variability of CCDC88b gene expression in lymphocytic subsets, these results should be formally presented.

Furthermore, the correlation analyses presented are not standard methodology in the field—given the extensive linkage disequilibrium (LD) in the region, the demonstration of modest correlation between the disease-associated and eQTL-associated SNPs is not convincing; I would be interested in seeing both a colocalization analysis (Plagnol *Biostatistics* 2009) that provides a posterior probability analysis that the lead disease-associated SNP is correlated with the lead eQTL, as well as multi-SNP transcriptome-wide analysis (Gusev *Nature Genetics* 2016) that incorporates LD patterns to provide multi-SNP estimates whether increased or decreased expression of cis-SNPs are significantly associated with the disease-association signal. Separate analyses of CD and UC would provide further insight here. I suspect that the TWAS association signal will demonstrate modestly-positive association of increased CCDC88b that will not be genome-wide significant, but given the compelling *in vivo* T cell transfer data, will provide a more statistically precise estimate of the eQTL significance here.

2) The other major limitation is assessing in what immune cells altered CCDC88b expression exerts its pathogenic role. While a more detailed *in vivo* interrogation in murine models is beyond the scope of this study, I would think that a more detailed human intestinal analyses, dissecting out in inflamed vs. non-inflamed, CD vs. UC, involving multi-parameter cellular analyses, should be provided here. Given the T cell transfer data, the eQTL differences in CD14+ cells, and the authors' speculation re the role of CCDC88b interactions with DOCK8 in NK cells (Ham et al., *J. Immune* 2015), I would favor a well-powered, systematic analysis of immune cell expression (to include these cells) in IBD tissue. The present immunohistochemistry and RT-PCR results add little mechanistic insight to the DSS induction of CCDC88b expression results. Ideally, these expression analyses might provide further insight as to whether the functional mechanisms include altered lymphocytic differentiation, enhanced chemokine secretion, and/or altered cytotoxicity.

June 15, 2017

Response to reviewers

RE: "CCDC88B IS REQUIRED FOR PATHOGENESIS OF INFLAMMATORY BOWEL DISEASE" NCOMMS-16-24887

We want to thank the reviewers for their time and their insightful comments and suggestions that certainly led us to improve our work. Please, find enclosed a revised version of our manuscript that has been amended according to the reviewer's suggestions and recommendations. In the letter below, we will explain how we addressed the reviewer's comments in the body of the revised text, including the addition of significant amount of new data that is included in some of the revised figures. Please note that all the changes in the manuscript are in red.

Reviewer #1 (Remarks to the Author):

Interesting study with both mice and human data making these findings potentially clinically relevant

1. Please use another model of experimental colitis (TNBS or oxazolone)

We thank the reviewer for highlighting the clinical relevance of our work. In our manuscript, we have tested the implication of Ccdc88b on gut inflammation in two independent mouse models, namely DSS-induced experimental colitis model and the more clinically relevant T cell transfer model. These models are commonly used for studying colitis induced by epithelial barrier damage and T cell transfer in Rag1^{-/-} mice, respectively. Repeating the same experiments using another model of experimental colitis such as TNBS or oxazolone, will at best only provide a small incremental amount of novel information at this stage, will require much technical effort, and goes beyond the scope of the current study.

Instead, we wanted to understand whether Ccdc88b expression is required in both innate and adaptive immune cells to promote pathogenesis during DSS-induced colitis. With this in mind, we generated Rag1^{-/-}Ccdc88b^{mut} mice that lack Ccdc88b as well as B and T adaptive immune cells. Our results clearly show that Rag1^{-/-}Ccdc88b^{mut} mice recovered from DSS-induced colitis when compared to Rag1^{-/-} mice suggesting that the absence of Ccdc88b expression by innate immune cells is sufficient to dampen epithelial barrier damage, decrease inflammation and protect against experimental colitis (please, see **supplementary Figure 2 and page 5 line 14**).

2. Methods: What is the reference of the primary and secondary antibodies used for detection of CCDC88B by WB?

The anti-CCDC88B rabbit hyperimmune antiserum was produced in our lab, and initially described in Kennedy, JM et al., *J. Exp. Med.* 211: 2519-35, 2014. The secondary antibody Goat anti-rabbit (Ref ab136636) was purchased from Abcam.

3. *Figure 1. Please perform some double immunostaining to investigate which type of T cells are involved here (CD4+ or CD8+)*

We thank the reviewer for his/her suggestion. We have performed double immunostaining for CCDC88B along with CD4 and CD8. These results now presented in **Figure 1e and page line 20** first demonstrate that both CD4+ and CD8+ T cells are recruited to the inflamed colon of mice treated with DSS (day 8) and that CCDC88B is expressed in both lymphoid and myeloid cells that infiltrate the inflamed colon of these mice. The expression pattern of CCDC88B in the inflamed colon could be attributed to the DSS colitis model we have used in these experiments, where myeloid cells play a predominant role. We have further validated the contribution of CCDC88B in innate immune myeloid cells during DSS-induced colitis by using the Rag1^{-/-} Ccdc88b^{mut} mice (please refer to our response above).

4. *Regarding KO mice, please show the expression levels of CCDC88B in mice colon.*

The *Ccdc88b* mutant mouse we are using in these studies carry a mutation in a splice site of intron 21 of the gene [Chr.19; T:C, position 6,922,670], which completely abrogates the normal splicing of the gene. The resulting truncated protein is unstable and not detected in normal tissues of the mutant animals [Kennedy, JM et al., *J. Exp. Med.* 211: 2519-35, 2014]. As requested by the reviewer, we have included an immunoblot performed on colon protein extracts from Wild-type and *Ccdc88b* mutant mice (**Figure 1b and page 4 line 10**). This result confirms the absence of mature CCDC88B protein expression in the colon tissue of mutant mice.

5. *Figure 6: Please investigate which type of cells are involved here similar to Figure 1*

As suggested by the reviewer, we conducted a series of experiments to characterize the expression of CCDC88B in different cell types involved in UC and CD pathogenesis. We carried out double immunofluorescence staining of CCDC88B, CD4 and CD68 (marker for myeloid cells) (**Figure 6d and e and page 7 line 8**) and single staining for CD3 and CD8 positive cells (**Supplementary Figure 5b and page 7 line 6**) in colon tissue sections derived from NL controls, CD or UC patients. We found increased infiltrations of CD3 and especially CD8 cell subsets in UC and CD specimens in comparison with NL controls (**Supplementary Figure 5b and page 7 line 6**). As we have shown in the mouse colon (**Figure 1c**), we saw very little expression of CCDC88B in NL colon tissue sections, but increased expression in colon tissue section of UC and CD patients (**Figure 6d and e**). Enhanced CCDC88B expression was accompanied with increased CD4⁺ and CD68⁺ cell infiltrate in colon tissues of UC and CD patients in comparison with NL controls. Interestingly, CCDC88B⁺ infiltrating cells were both CD4⁺ and CD68⁺ cells with a co-localization of CCDC88B with both subsets (**Figure 6d and e and page 7 line 8**).

6. *Human data: please show the correlation between inflammatory markers and expression levels of CCDC88B*

We truly appreciate the reviewer's suggestion to show correlation between inflammatory markers and levels of CCDC88B in human tissue samples. We performed qPCR analysis of IL-8 expression, a key pro-inflammatory cytokine, in primary colon tissue from NL control individuals, CD and UC patients. IL-8 mRNA expression was profiled in 7 CD patients, 38 UC patients, and 69 NL controls. We noted a significant increase in IL-8

mRNA levels in UC and CD patient samples compared to NL controls, which correlated with the expression levels of CCDC88B (**Figure 6c and page 6 line 42**).

7. *Human data: colorectal cancer or diverticulitis is not a good control. Please use non-inflamed areas (macroscopically and histologically) from colon of IBD patients.*

We thank the reviewer for this suggestion and agree with him/her that it would be ideal to use non-inflamed matched control tissues from the same IBD patient and compare these to the inflamed tissues. However, this is almost impossible to obtain as IBD patients are already undergoing surgery for inflamed bowel removal and there is no guarantee that the adjacent tissue (>10 cm below or above the resected inflamed IBD area) does not contain infiltrates of inflammatory immune cells. Using non-inflamed tissue samples adjacent (>10 cm) to the resected colorectal cancer or diverticulitis area is common in the field when comparing these to colon tissue specimens obtained from IBD patients (please refer to our collaborators work *Gastroenterology*. 2011 Feb;140(2):550-9; *Gastroenterology*. 2013 Mar;144(3):601-612.e1).

To confirm the inflamed nature of the tissue samples used in our experiments, hematoxylin & eosin (H&E) staining was performed on paraffin slides originated from the same cohort used in this study and histological scoring was assessed blindly by a pathologist. In **Supplementary Figure 5 and page 7 line 2**, we now confirm that the colonic mucosa of NL controls has no significant histological changes and no inflammation, however, the colonic mucosa of CD and UC patients presents with increased chronic inflammation, gland distortion as well as active inflammation. In addition, these results further validate our observation that high levels of the pro-inflammatory cytokine IL-8 correlate with increased inflammation in the colon tissue of UC and CD patients when compared to NL control tissue specimens (**Figure 6c and page 6 line 42**).

8. *Figure 4: Please look at expression levels of pro- and anti-inflammatory cytokines within colon*

We assessed the mRNA expression of pro-inflammatory cytokines in the colon of mice following CD4⁺ T cell transfer. Overall, we have seen a trend but not significant increase of TNF, and IFN γ in colon of *RAG1*^{-/-} mice transferred with WT CD4Rb^{hi} than *Ccdc88b*^{mut} CD4Rb^{hi}.

9. *Methods: please give further details on the histological score used for animal experiments*

Further details on the calculation of the histological score is now included as **supplementary Table 1**.

Reviewer #2 (Remarks to the Author):

*This study examines the effects of CCDC88b on two models of intestinal inflammation, namely the DSS and T cell transfer model. They demonstrate that in both models, mutant CCDC88b protects against colitis. In the DSS model, they demonstrate an induction of CCDC88b expression. This group's interest in this molecule stems from their prior work (Kennedy et al., *JExpMed* 2014) using an ENU screen to identify factors that protect against cerebral malaria. In that prior work, they demonstrated that CCDC88b mutants demonstrate impaired T*

cell maturation/differentiation. *CCDC88b* is expressed in both lymphocytic and myeloid cell populations. They also show data that from PBMCs from 320 individuals that there is an eQTL in *CD14+* peripheral blood monocytes that correlates with the maximally disease-associated SNPs. Finally, they show expression data from IBD intestine demonstrating induction of *CCDC88B* in non-epithelial cells of the colon from both CD and UC.

The strengths of this study include the relative novelty of this understudied protein and the compelling role that *CCDC88b* plays in the T cell transfer model of colitis. Given the paucity of functional analyses of IBD-associated loci, this study represents a much needed functional interrogation of disease-associated loci in IBD.

The major weaknesses/interpretative limitations of the study include

1) The eQTL analysis of peripheral blood (Figure 5). The analyses provided are limited due to the fact that the authors only present the results from *CD14+* cells, despite having generated data from *CD4+*, *CD8+*, *CD19+* and *CD15+* cells. Even if negative, and even if the authors don't see compelling inter-individual variability of *CCDC88b* gene expression in lymphocytic subsets, these results should be formally presented.

As asked by the reviewer, we are providing the analysis in **supplementary Figure 3b left and page 6 line 20** of the effect of different SNP at the 11q13 locus on *CCDC88B* mRNA expression in *CD4+*, *CD8+*, *CD15+* and *CD19+* cells. The corresponding top SNPs and corresponding eQTL p values are , rs61886930; p=0.0001 for *CD4+*, rs475688 ; p=0.02513 for *CD8+*, rs10751006 ; p=0.0004 for *CD15+* and rs7119252 ; p=0.002265 for *CD19+*. However, we found no significant correlation between eQTL effects on *CCDC88B* mRNA expression and association with CD GWAS in these cell subsets **supplementary Figure 3b right and page 6 line 20**. The only positive and significant correlation is the one we present in the body of the paper for *CD14+* cells.

Furthermore, the correlation analyses presented are not standard methodology in the field—given the extensive linkage disequilibrium (LD) in the region, the demonstration of modest correlation between the disease-associated and eQTL-associated SNPs is not convincing; I would be interested in seeing both a colocalization analysis (Plagnol Biostatistics 2009) that provides a posterior probability analysis that the lead disease-associated SNP is correlated with the lead eQTL, as well as multi-SNP transcriptome-wide analysis (Gusev Nature Genetics 2016) that incorporates LD patterns to provide multi-SNP estimates whether increased or decreased expression of cis-SNPs are significantly associated with the disease-association signal. Separate analyses of CD and UC would provide further insight here. I suspect that the TWAS association signal will demonstrate modestly-positive association of increased *CCDC88b* that will not be genome-wide significant, but given the compelling in vivo T cell transfer data, will provide a more statistically precise estimate of the eQTL significance here.

We thank the reviewer for his/her comments. We have used an alternate method to evaluate whether eQTL and GWAS association patterns are similar. We elected to use the SMR method developed by Peter Visscher's group (Zhu et al. Nature Genetics 48: 481-487, 2016). It confirms the initial results obtained with our method where the correlation p value between of the top eQTL *CCDC88B* mRNA expression SNP rs538147 and GWAS dataset is 0,01 in *CD14+* cells. In the SMR approach, we found a positive and significant correlation only in *CD14+* cells (0.014) associated with a high p-value for the HEIDI test (0.267) indicative of pleiotropy/causality.

2) The other major limitation is assessing in what immune cells altered *CCDC88b* expression exerts its pathogenic role. While a more detailed in vivo interrogation in murine models is beyond the scope of this study,

I would think that a more detailed human intestinal analyses, dissecting out in inflamed vs. non-inflamed, CD vs. UC, involving multi-parameter cellular analyses, should be provided here. Given the T cell transfer data, the eQTL differences in CD14+ cells, and the authors' speculation re the role of CCDC88b interactions with DOCK8 in NK cells (Ham et al., J. Immune 2015), I would favor a well-powered, systematic analysis of immune cell expression (to include these cells) in IBD tissue. The present immunohistochemistry and RT-PCR results add little mechanistic insight to the DSS induction of CCDC88b expression results. Ideally, these expression analyses might provide further insight as to whether the functional mechanisms include altered lymphocytic differentiation, enhanced chemokine secretion, and/or altered cytotoxicity.

We thank the reviewer for the suggestion and agree that multi-parameter staining of cells derived from inflamed and non-inflamed colon tissue from CD and UC patients will be best. Unfortunately, the anti-human CCDC88B antibody produced by our laboratory works only for western blotting and immunohistochemistry/fluorescence at this stage. Despite our efforts to enhance the recognition and specificity of this antibody, we were not able to successfully apply it for flow cytometry. In addition, the commercially available sources of anti-human CCDC88B antibodies are only validated and used for western blotting and immunohistochemistry/fluorescence applications. Nevertheless, to respond to the reviewer's critique, we provide evidence that CCDC88B is highly expressed in the colon of CD and UC patients when compared to NL controls and that CCDC88B⁺ infiltrating cells are both CD4⁺ and CD68⁺ cells. For a detailed description of these results, please refer to **Figure 6d and 6e and page 7 line 8**, as well as our response to reviewer 1 (point #5).

As suggested by the reviewer and using Flow cytometry analysis, we have profiled the cell types that express Ccdc88b and are infiltrating into the inflamed colons of mice at day 0, day 4, and day 8 following the treatment with DSS. Our results presented in **Figure 1d and page 4 line 16 and supplementary Figure 1 and page 4 line 22**, clearly show the infiltration of CD4 T, CD8 T, NK cells, neutrophils, inflammatory monocytes and macrophages Ccdc88b⁺ CD45⁺ myeloid and lymphoid cells in the colon of mice treated with DSS on days 4 and 8 compared to day 0.

REVIEWERS' COMMENTS:

Reviewer #1 (Remarks to the Author):

The authors adequately and extensively responded to my comments

Reviewer #2 (Remarks to the Author):

In this re-submission, the authors provide the absence of eQTL significance in cells besides the positive CD14+--As a reader, I would like for the authors to explain in the conclusion section some reconciliation between the eQTL studies implicating increased expression in only CD14+ cells, with their findings regarding the necessity of CCDC88b for colitis in the RAG/lymphocyte deficient reconstitution model. This will, of necessity, be a bit speculative in nature.

The added immunohistochemistry provides a larger contextual significance to CCDC88b expression in inflamed CD and UC. The broad expression profile and the induction in both CD and UC are consistent with the DSS data that opens the paper.

REVIEWERS' COMMENTS:

Reviewer #1 (Remarks to the Author):

The authors adequately and extensively responded to my comments

Reviewer #2 (Remarks to the Author):

In this re-submission, the authors provide the absence of eQTL significance in cells besides the positive CD14+/-As a reader, I would like for the authors to explain in the conclusion section some reconciliation between the eQTL studies implicating increased expression in only CD14+ cells, with their findings regarding the necessity of CCDC88b for colitis in the RAG/lymphocyte deficient reconstitution model. This will, of necessity, be a bit speculative in nature.

To address the reviewer comment, we have added the following statement to the Discussion of the paper:

“The specificity of the eQTL effect on CCDC88B expression in human CD14+ myeloid cells and its association with IBD disease risk may seem in apparent contrast to the results obtained in the RAG/lymphocyte deficient reconstitution model and that support a role of CCDC88b in T cells in this mouse model of intestinal inflammation. We believe this points to a key role of CCDC88B in both lymphoid and myeloid cells, with the former being critical in this mouse model of inflammation, while the latter is possibly more relevant to the human disease and the criteria used to establish diagnosis. Indeed, we have previously demonstrated a role for CCDC88B in the function of both T cells and myeloid cells in other mouse models of inflammation (neuroinflammation)⁵.”

The added immunohistochemistry provides a larger contextual significance to CCDC88b expression in inflamed CD and UC. The broad expression profile and the induction in both CD and UC are consistent with the DSS data that opens the paper.